# Reduction of aerosol dissemination in a dental area generated by high-speed and scaler ultrasonic devices employing the "Prime Protector"

**Esthelvia Carolina Guzmán-Flores**(ID)*[�‡], **Amparo Rocío Fuentes-Ayala**[�‡], **Alicia Consuelo Martínez-Martínez**[�‡], **Daniela Estefanía Aguayo-Félix**[�‡], **Margarita Valeria Arellano-Osorio**[�‡], **Martín Campuzano-Donoso**(ID)[�‡], **Náthaly Mercedes Román-Galeano**(ID)[�‡], **Melanie Llerena-Velásquez**[�‡], **Yajaira Vásquez-Tenorio**[�‡]

School of Dentistry, Faculty of Medical Sciences, International University of Ecuador, Quito, Ecuador

☯ These authors contributed equally to this work.

* esguzmanfl@uide.edu.ec

**Data Availability Statement:** The minimal data set necessary to replicate this study's findings is

## Abstract

The use of an external dome aerosol containment device (Prime Protector) is proposed to reduce the spread of particles within the dental office. Hence, the aim of our study was to compare the spread of bioaerosols generated by a High-speed Handpiece (HH) and an Ultrasonic Prophylaxis Device (UPD), with and without the Prime Protector dome (PP) by counting Colony Forming Units (CFU) of Lactobacillus casei Shirota, at different distances on the x and y axis. The PP was located considering the parallelism between the base of the dome and the frontal plane of the simulator, aligning the center of the mouth with the center of the dome. The PP dome measurements are 560.0mm x 255.0mm x 5mm. Petri dishes were placed at 0.5 m, 1 m and 1.5 m respectively. Aerosol generation in the laboratory environment was done three times with the following experimental groups 1) HH, 2) HH-PP, 3) UPD, 4) UPD-PP. Each dental device activation (HH and UPD) had a time frame of 2 minutes on the upper anterior teeth of the dental phantom with a liquid suspension containing Lactobacillus casei Shirota (YAKULT 0836A 0123; 1027F 0407). Air pressure and ventilation were parameterized. No separate high-volume evacuation used, nor was there any air removal attached to the dome. Results showed no significant difference between distance and axis in the CFU count. When means for devices and distances were compared between each of them all showed significant differences except for UPD and UPD-PP (p <0,004). In conclusion, external devices like Prime Protector could help decrease aerosol diffusion during high-speed handpiece activation. However, this dome does not replace the use of PPE inside dental clinics.

## Introduction

Since the beginning of the COVID-19 pandemic, biosafety measures in dental offices have been reinforced by seeking new alternatives to protect patients' health. In addition, the update

available from Figshare at DOI:10.6084/m9.
figshare.23628990 (https://doi.org/10.6084/m9.
figshare.23628990.v1).

**Funding:** The author(s) received no specific
funding for this work.

**Competing interests:** The authors have declared
that no competing interests exist.

of protocols has raised questions regarding biosafety specifically, about the reduction of aerosols with biological load emitted during dental practice. This represents a potential threat to clinicians and assistants [1, 2].

Within the current context of the pandemic, it is known that the main routes of transmission for SARS-CoV-2 include inhalation of droplets and aerosols, and direct contact of the conjunctival nasal or oral mucosa with contaminated body fluids [3]. However, this virus is not the only potential threat in dental practice. Pathogenic microorganisms native to the human respiratory tract can contaminate the environment and surfaces through suspended particles resulting from Aerosol Generating Procedures (AGP) representing a risk of contracting diseases such as influenza, meningitis, tuberculosis, among others [4].

According to Matys & Grzech-Leśniak (2020), there are three types of aerosols generated during dental practice. The first type is produced through breathing, sneezing and coughing. The second is generated purely by rotary instruments and the third is a mixture between spray aerosol and respiratory bioaerosols [5]. These particles are less than 5 micrometers in size and remain suspended in the environment even after the treatment has ended, representing a biological respiratory hazard to professionals and patients [6, 7].

To mitigate this risk, the CDC (Centers for Disease Control and Prevention) presents a guide that determines the hierarchy of control measures for worker hazards, according to their effectiveness composed by 5 levels which are Elimination, Substitution, Engineering controls, Administrative controls and Personal protective equipment (PPE). As a result of the pandemic, emphasis was placed on the development of engineering controls in the dental field, to find a barrier that protects both dental operators and patients from AGP [8].

Among the devices mentioned in the literature it is possible to find a plastic-coated metal frame, a protection chamber, a dome-shaped customized shield and some other aerosol containment devices [1, 9, 10]. Some of these physical barriers are associated with high suction systems, such as the one proposed by Teichert-Filho et al. in 2020. and Suwandi et al. in 2022 [11, 12] while others do not consider this variable [1, 9, 13].

In addition, high-powered dental suction, rubber dam, ventilation and antimicrobial mouth rinses have shown an effective reduction of aerosol and droplets contamination in the dental office [14]. For this reason, the ADA (American Dental Association) recommends the use of these strategies.

Recent studies aiming to reduce the aerosol dispersion show a variety of results because of different methods, devices and variables considered [5, 12]. This concocts contrasting ideas on how to improve biosecurity through innovation in personal protective equipment (PPE), high-power suction or aerosol containment equipment. [1–3].

In the present study, the use of an external dome aerosol containment device (Prime Protector) is proposed to reduce the spread of particles within the dental office without considering any attached air removal equipment [12]. Hence, the aim of our study was to compare the spread of bioaerosols generated by a High-speed Handpiece (HH) and an Ultrasonic Prophylaxis Device (UPD), with and without the Prime Protector dome (PP) by counting Colony Forming Units (CFU) of Lactobacillus casei Shirota, at different distances on the x and y axis.

## Materials and methods

The methodology used in this study is based on a previous work in which Lactobacillus casei Shirota is used as a biological marker of aerosol dispersion [15]. However, the authors made variations regarding the aerosol containment device, the activation time of the handpieces, the exposure time of the petri dishes to the aerosols, the facilities where the experiment was carried out, and the use of an oral cavity simulator.

## Standard experimental conditions

This investigation took place at the Microbiological and Preclinical laboratories from the International University of Ecuador, Dentistry School. Inside the microbiological laboratory the Petri dishes (94 mm x 16 mm) were prepared containing an enriched medium for Lactobacillus spp. (Lactobacillus MRS Agar, TM MEDIA, M1E6FV01). The preclinical laboratory where the study was conducted has 15 dental-chairs simulators and has a working area of 6.14 m x 10.23 m x 8.21 m. The experiment was run during the summer break to avoid any possible cross contamination (Fig 1A).

The head of the dental-chair simulator was used as reference to set up tables at a similar height to dental auxiliary tables (0,76 m) in a vertical and horizontal axis where the Petri dishes were placed at 0.5 m, 1 m and 1.5 m respectively (Fig 1B). Foregoing the activation of the high-speed devices the working area was disinfected, and temperature (20˚C) and humidity (64%) were registered. A control Petri dish was placed in the Preclinical laboratory once the table, floor, and dome cleaning and disinfection processes were completed, until the next experiment started. This was done to discard crossed contamination from the previous experiment.

Furthermore, the air and ventilation conditions were parameterized meaning that windows, doors, and air circulators were closed during the experimental phase to avoid intervention with the aerodynamic air circulation inside the laboratory. Finally, in order to dismiss any possible environmental contamination, a Petri dish containing MRS was left inside the laboratory for 20 minutes prior to the beginning of the experiment. This protocol was repeated every time.

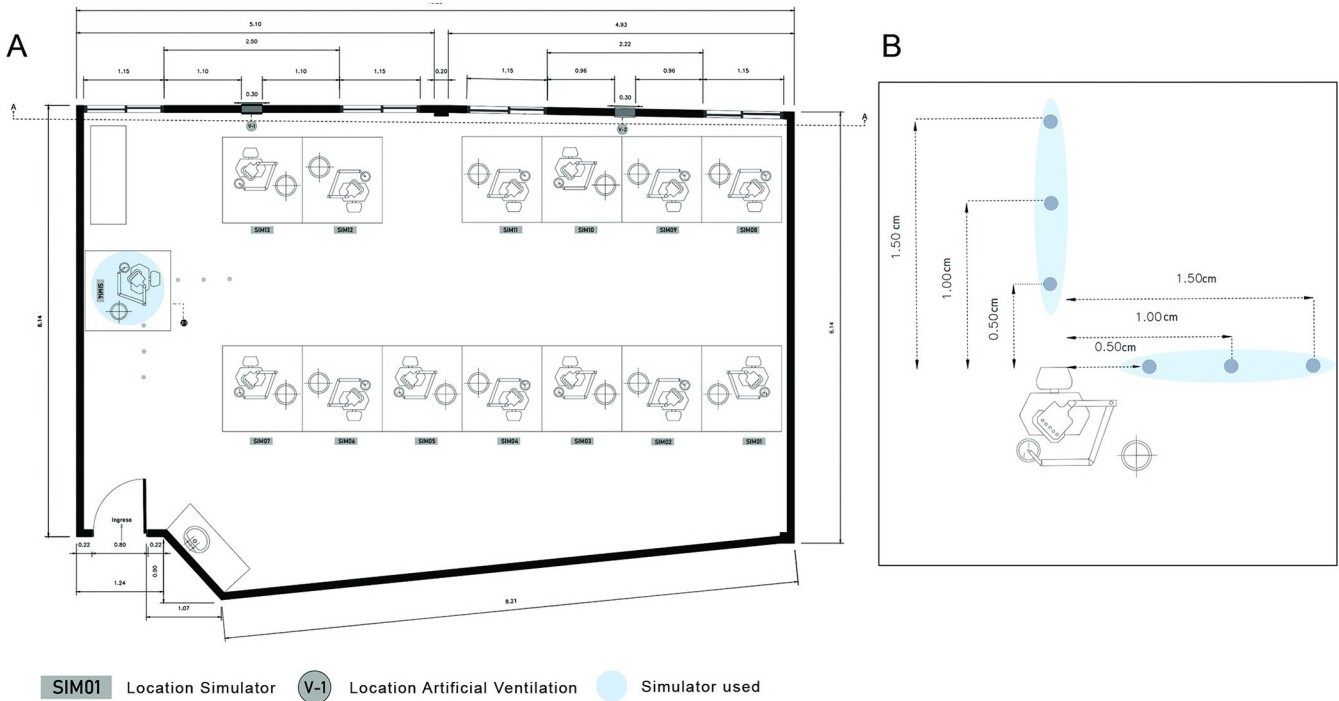

SIM01 Location Simulator    V-1 Location Artificial Ventilation    Simulator used

**Fig 1.** A) Preclinical laboratory distribution, floor plan. Simulator 014 was used as shown and highlighted in blue B) Petri dishes schematic distribution from the dental chair headboard.

## Microorganism used

To fill up the water container from the simulator unit (which was previously disinfected with alcohol 70%) a liquid suspension was prepared containing Lactobacillus casei Shirota (YAKULT 0836A 0123; 1027F 0407) at a 0.5 concentration according to McFarland's scale. Lactobacillus has a similar size proportion as oral bacteria, besides being innocuous to the environment and human life. By virtue of this last characteristic, it was not necessary to get an authorization from the Environmental and Health Department nor the Bioethical Committee.

## Experimental phase

To emulate the oral cavity a dental phantom was used (OM-860-1) at the simulator's head where the ultrasonic device (Dentflex Cavflex 6000) with a 01 Tip Perio E and handpiece (NSK PAP-SU B2) with a FG1012 bur were activated. Aerosol generation in the laboratory environment was done three times with the following experimental groups 1) HH, 2) HH-PP, 3) UPD, 4) UPD-PP. Each dental device activation (HH and UPD) had a time frame of 2 minutes on the upper anterior teeth of the dental phantom with water and air pressure of 39 ± 2 PSI and a water flow rate of 160± 3 mL/min. Each activation was performed by the same operator (1.59 m). Once the HH and UPD were deactivated, all Petri dishes were opened and after 30 minutes exposure to the environment they were closed. Afterwards the Petri dishes were placed in an aerobic incubator (MEMMERT, GmbH+ CO. KG, D-91126 Schwabach FRG, Germany) for 48 hours at 35 ± 2˚C.

During this phase an external aerosol containing device (Prime Protector) was used (Fig 2). The device has two main parts: a transparent methacrylic (PMMA) dome and a 304 stainless-steel structure. The PMMA dome weighs 0.6kg and its measurements are 560.0mm x 255.0 x 5 mm. The dimensions of the stainless-steel structure were 9444,0mm x 570,0mm x 1250,0 mm and a weight of 20kg. This device has a stainless-steel base with three feet joined by a central axis. The PP was located considering the parallelism between the base of the dome and the frontal plane of the simulator, aligning the center of the mouth with the center of the dome. The PP dome measurements are 560.0mm x 255.0mm x 5mm.Once the experimental phase concluded, all doors, windows and air systems were opened to allow air flow.

## Microbiological analysis

The CFUs were manually counted by the authors who were previously trained by a specialist in microbiology, and the results were verified by the same expert. Furthermore, the presence of Lactobacillus casei Shirota was confirmed through Gram stain analysis.

## Statistical analysis

Data from the experiments were examined for normality by the Shapiro Wild test. Once the data was verified, the non-parametric test Kruskall-Wallis was applied. Post hoc analyses used the MannWhitney U multiple comparisons criterion for distances and axes, obtaining significance at 95%. All the statistical tests were performed on the Statistical Package for Social Sciences (SPSS).

# Results

The unit of analysis were the CFUs counted in the different established groups: distance (0.5 m; 1 m and 1.5 m), location (x, y axis) and device (HH, HH-PP, UPD, UPD-PP). The highest CFU count was found at HH with a mean and standard deviation of 2.943,29 and 515.01 respectively. Followed by the 0.5 m distance with a mean and standard deviation of 1.013,63

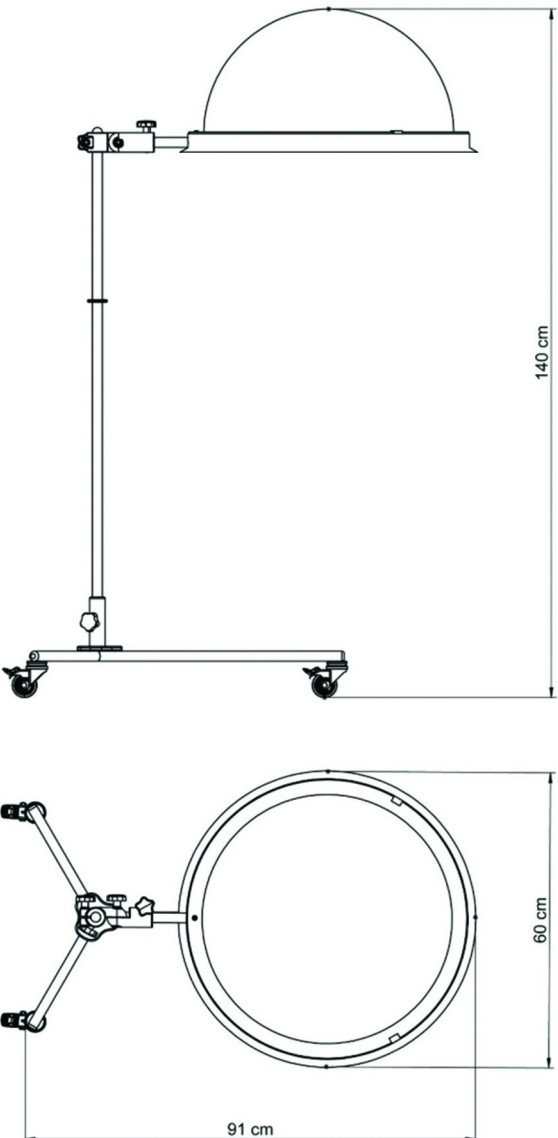

**Fig 2. Transparent methacrylic dome Prime Protector.**

and 1.362,04. Otherwise, the lowest CFU count was at UPD with a mean of 10.56 and standard deviation of 6.4. Fig 3 shows the established groups distribution in order to know the behavior of the CFUs, denoting that the device marks a difference between them. The HH generated a superior CFU count out of all the devices. One of the Petri dishes located at 1.5 m was invalidated because of a traceability break.

To identify the statistical difference in CFU count for each group a mean comparison was made resulting in a parametric ANOVA test for the devices and a non-parametric test Kruskall-Wallis for the axis and distance variables. The tests were run at a 95% trust level and a significant difference was found between the devices (HH, HH-PP, UPD, UPD-PP); F (3) = 381.028; p = 0.000. Between distance and axis there was no significant difference in the CFU count.

Post Hoc analysis used Tukey multiple comparisons when comparing devices and Mann-Whitney U test for significance showed that the main difference was between HH and HH-PP,

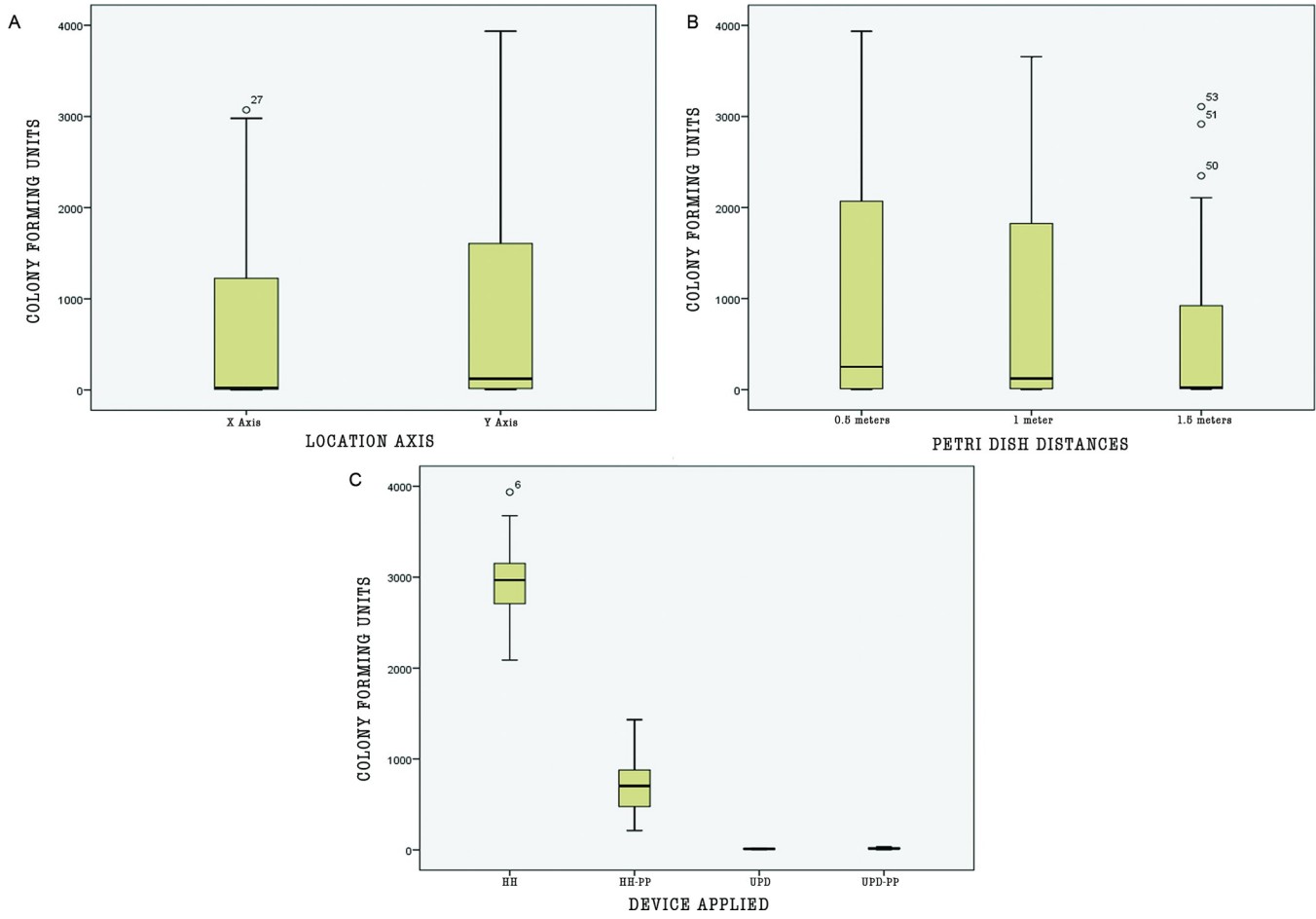

**Fig 3. Box plot of the data of CFU counts versus location, distance and device.** A) Colony Forming Units per axis location, B) Colony Forming Units per each distance, C) Colony Forming Units per device. Mean differences between groups with Kruskall Wallis and post hoc using Mann-Whitney U multiple comparison test.

HH and UPD, HH and UPD-PP, HH-PP and UPD, HH-PP and UPD-PP (p<0.0001) meanwhile UPD and UPD-PP did not show significant difference. The mean difference between HH and HH-PP was 75.9%, HH and UPD 99.6% and HH and UPD-PP 99.4%, finally UPD and UPD-PP mean was 35.8%.

When means for devices and distances were compared between each of them all showed significant differences except for UPD and UPD-PP (p <0,004) For UPD-PP CFUs recount showed a significant difference between the petri dishes distances. (x, y axis) (p = 0,013) (Table 1).

## Discussion

A considerable number of microorganisms are present in the oral cavity, some of them with pathogenic potential [10, 11]. Therefore, arises the need to develop studies that guide dental professionals in the application of effective ways to minimize the exposure of dental workers. The hierarchy of controls by CDC helps determine which actions can better control exposures to hazards, based on its five levels of effectiveness.

**Table 1. Mean differences between each distance and device group.**

| Devices/ Axis | 0.5 m | | | 1 m | | | 1.5 m | | |
|---|---|---|---|---|---|---|---|---|---|
| | N | Mann-Whitney U | Asymptotic sig. (bilateral) | N | Mann-Whitney U | Asymptotic sig. (bilateral) | N | Mann-Whitney U | Asymptotic sig. (bilateral) |
| HH–HHPP | 12 | 0.000 | 0.004* | 12 | 0.000 | 0.004* | 11 | 0.000 | 0.006* |
| HH–UPD | 12 | 0.000 | 0.004* | 12 | 0.000 | 0.004* | 11 | 0.000 | 0.006* |
| HH—UPDPP | 12 | 0.000 | 0.004* | 12 | 0.000 | 0.004* | 11 | 0.000 | 0.006* |
| HHPP–UPD | 12 | 0.000 | 0.004* | 12 | 0.000 | 0.004* | 12 | 0.000 | 0.004* |
| HHPP–UPDPP | 12 | 0.000 | 0.004* | 12 | 0.000 | 0.004* | 12 | 0.000 | 0.004* |
| UPD–UPDPP | 12 | 11.000 | 0.261 | 12 | 15.000 | 0.631 | 12 | 13.000 | 0.418 |
| X axis–Y axis | 24 | 56.000 | 0.356 | 24 | 60.000 | 0.488 | 23 | 53.500 | 0.441 |

HH, High-speed Handpiece; HHPP, High-speed Handpiece with the Prime Protector; UPD, Ultrasonic Prophylaxis Device; UPDPP, Ultrasonic Prophylaxis Device with the Prime Protector.

* Represents significance ($p < 0.05$). Mean differences between groups with Kruskall Wallis and post hoc using Mann-Whitney U multiple comparison test.

The most effective is removal of the hazardous agent from the source, followed by the substitution with safer alternatives that minimize exposure risks. The application of these first two levels was not the aim of this study.

Within the third level are Engineering Controls, aimed to reduce or prevent danger by preventing contact with the operators. In the dental clinical area these can be equipment modifications, ventilation, reduction of the volume of water used during dental procedures could reduce the aerosol at speeds below 100.000 rpm [16, 17] and use of protective barriers, among which is the PP. Likewise, a study published by Montalli et al in 2021 demonstrated that the use of an aerosol control device on the ultrasound tip significantly reduced the emitted aerosols [1].

Data demonstrates that the use of multiple mitigation strategies such as high-power dental suction, rubber dam, ventilation, and the novel aerosol containment devices like a metal frame with plastic wrap and a plastic shield chamber reduce the contamination [1, 9, 10, 17]. Our study evaluates the efficacy of the Prime Protector external device for containment of aerosols.

Finally, administrative controls and PPE (fourth and fifth levels respectively) establish work practices that reduce the duration, frequency or intensity of exposure. Among them are training work processes, rotations, limiting access to dangerous areas. Several clinical studies have now shown that the combination of preprocedural mouth rinsing and the use of high-volume evacuation allows dental practice to be conducted more safely [14, 18].

Being the last levels of risk prevention, it is not appropriate to use them as the only protection methods, especially when there are other options to reduce it [19] Dental practice can be executed safely when adequate control measures and biosecurity protocols are applied, such as the proper use of surface disinfection and PPE, which are essential to avoid cross-infection [20, 21].

It is important to emphasize that despite the number of studies that exist on the dispersion of aerosols regarding the engineering control alternatives in the dental office, results vary depending on various factors including procedures, equipment used and the condition under each study is carried out [22]. Taking these factors into consideration, the present study sought to incorporate different variables such as distances, devices, and the use of Prime Protector.

Several reports show that aerosol contamination can reach 1.15 m [23], 1.2 m [24], 1.5 m [22], 2 m [25] and up to 3 m [9]. Kaufmann et al., [25] mention that the maximum aerosol

contamination is at 0.7 m from the patient's mouth, while Gallagher et al., [22] state that it is between 0.2 and 0.5 m. According to the results of our study, a significantly greater presence of CFUs was found at the shortest distance (0.5m) in relation to the operative field (or simulator's head) than at 1m and 1.5m with and without the use of the Prime Protector external device. Although the amount of CFU counted was lower when using the Primer Protector device, the aerosol particles managed to reach the furthest Petri dish.

Other studies [26, 27] have observed the scope of the particles that emanate from the most used devices in dental practice such as HH and UPD. According to our results, the average difference between HH and UPD reaches 99.6%, that means UPD generates almost no aerosol compared to the use of HH. These results coincide with other existing data [28]. This could be related to the size of the particles produced by the instruments. In the case of HH, the particles are smaller and therefore reach a greater distance than the large particles generated by the UPD.

The external barrier PP used for this study mitigated the aerosol dispersion generated by HH to 75,9%. In a similar study [9], two external barriers showed as in our results a significant reduction in aerosol dispersion. However, as in our study there was still contamination on the surface even if it was lesser than when the devices were not used. Therefore, using PP does not completely ease aerosol dispersion. Nevertheless, PP only reduces aerosol dispersion by 35.8% when used with UPD. If compared to Montalli et all., [1] their device decreased aerosol dispersion by 98%. This could be related to device design, hence the big difference in aerosol dispersion.

For this study the microorganism used was Lactobacillus casei Shirota, bacillus, facultative aerotolerant anaerobe and with probiotic and innocuous characteristics for humans [29]. Moreover, an enriched culture medium M.R.S Agar for lactobacillus was used to cultivate L. casei Shirota since it contains high levels of carbon, nitrogen and other elements needed for its growth, this providing more reliable results. The reason behind selecting this specific microorganism has to do with its comparable size with various viruses and bacteria responsible for infectious diseases to which dental personnel are exposed.

On account of all the various microorganisms we are constantly exposed to, it is imperative to evaluate devises or implements in order to grant dentists more security while attending critical patients that require more urgent and elective treatments.

The methods employed in this study were designed to prove if the containing external device Primer Protector decreases aerosol dissemination spread. Because of this the authors worked under maximum aerosol generation conditions which resulted in a significant reduction of aerosols when the PP was used. Notwithstanding, working under real conditions with a patient the results could be different hence, some experimental circumstances should be considered as a limiting factor.

The activation time of the handheld devices, which is not directly comparable to the actual working time spent on dental procedures. This may mean a lower level of contamination than normal. The non-use of suction during the clinical trial is another limitation, since it has been shown that any type of suction, whether intraoral or extraoral, mitigates the spread of bioaerosols during some dental procedures.

Some recommendations the authors think should be acknowledged for future studies are: 1) real timing when performing a dental treatment, 2) use of a low-speed saliva ejector, 3) the inclusion of high-volume evacuation, 4) settling time of bioaerosols 5) patient's perception regarding dental treatment using physical barriers as the PP. Furthermore, it would be important to regard the patients' and dentists' perception about using external devices similar to the one used in this study.

## Conclusions

Our study shows that external devices like Prime Protector could help decrease aerosol diffusion during high-speed handpiece activation.

## Supporting information

**S1 File.**
(DOCX)

## Acknowledgments

The Prime Protector device was loaned to the authors for local evaluation by Johann Pérez CEO_Prime Health Protection S.L. Valencia—Spain. A special thanks to architect Lissette Lucio, who provided us with the laboratory plan layout at no charge. The authors received no financial support.

## Author Contributions

**Conceptualization:** Esthelvia Carolina Guzmán-Flores, Amparo Rocío Fuentes-Ayala, Alicia Consuelo Martínez-Martínez.

**Data curation:** Yajaira Vásquez-Tenorio.

**Formal analysis:** Yajaira Vásquez-Tenorio.

**Funding acquisition:** Esthelvia Carolina Guzmán-Flores.

**Investigation:** Esthelvia Carolina Guzmán-Flores, Amparo Rocío Fuentes-Ayala, Alicia Consuelo Martínez-Martínez, Daniela Estefanía Aguayo-Félix, Margarita Valeria Arellano-Osorio, Martín Campuzano-Donoso, Náthaly Mercedes Román-Galeano, Melanie Llerena-Velásquez.

**Methodology:** Esthelvia Carolina Guzmán-Flores, Amparo Rocío Fuentes-Ayala, Alicia Consuelo Martínez-Martínez.

**Project administration:** Esthelvia Carolina Guzmán-Flores.

**Resources:** Esthelvia Carolina Guzmán-Flores, Amparo Rocío Fuentes-Ayala, Alicia Consuelo Martínez-Martínez.

**Supervision:** Esthelvia Carolina Guzmán-Flores.

**Validation:** Yajaira Vásquez-Tenorio.

**Visualization:** Martín Campuzano-Donoso, Náthaly Mercedes Román-Galeano.

**Writing – original draft:** Esthelvia Carolina Guzmán-Flores, Amparo Rocío Fuentes-Ayala, Alicia Consuelo Martínez-Martínez, Daniela Estefanía Aguayo-Félix, Margarita Valeria Arellano-Osorio, Martín Campuzano-Donoso, Náthaly Mercedes Román-Galeano, Melanie Llerena-Velásquez.

**Writing – review & editing:** Esthelvia Carolina Guzmán-Flores, Martín Campuzano-Donoso, Náthaly Mercedes Román-Galeano, Melanie Llerena-Velásquez.

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
