## [Decision Letter · Decision Letter 0]

24 Jan 2023

PONE-D-22-32218Reduction of aerosol dissemination in a dental area generated by high-speed and scaler ultrasonic devices employing the “Prime Protector"PLOS ONE

Dear Dr. Guzmán Flores,

Thank you for submitting your manuscript to PLOS ONE. After careful consideration, we feel that it has merit but does not fully meet PLOS ONE’s publication criteria as it currently stands. Therefore, we invite you to submit a revised version of the manuscript that addresses the points raised during the review process.

We look forward to receiving your revised manuscript.

Kind regards,

Rajeev Singh

Academic Editor

PLOS ONE

*Comments from PLOS Editorial Office: We note that one or more reviewers has recommended that you cite specific previously published works. As always, we recommend that you please review and evaluate the requested works to determine whether they are relevant and should be cited. It is not a requirement to cite these works. We appreciate your attention to this request.*

Journal Requirements:

2. We note that Figure 1 in your submission contain copyrighted images. All PLOS content is published under the Creative Commons Attribution License (CC BY 4.0), which means that the manuscript, images, and Supporting Information files will be freely available online, and any third party is permitted to access, download, copy, distribute, and use these materials in any way, even commercially, with proper attribution. For more information, see our copyright guidelines: http://journals.plos.org/plosone/s/licenses-and-copyright.

Reviewers' comments:

Reviewer's Responses to Questions

**Comments to the Author**

1. Is the manuscript technically sound, and do the data support the conclusions?

Reviewer #1: Partly

Reviewer #2: Partly

2. Has the statistical analysis been performed appropriately and rigorously? 

Reviewer #1: No

Reviewer #2: Yes

3. Have the authors made all data underlying the findings in their manuscript fully available?

Reviewer #1: No

Reviewer #2: No

4. Is the manuscript presented in an intelligible fashion and written in standard English?

Reviewer #1: Yes

Reviewer #2: Yes

5. Review Comments to the Author

Reviewer #1: This is an interesting paper, and adds to the literature on accessory devices for assisting with improving air quality in the dental operatory.

The following revisions are necessary.

1.For the abstract to be meaningful, it should state that there was no separate high-volume evacuation used, nor was there any air removal attached to the dome. It should also give the key dimensions of the dome such as its diameter and position in relation to the mouth.

2.On line 56, the accepted definition for aerosol particles is a size of less than 5 µm - not 50 µm.

3.In the introduction on line 66, there is no mention of other effective risk reduction methods such as preprocedural use of antimicrobial mouth rinses.

4.In line 68 some discussion about the use of the dome would be useful. Without any connected air removal the dome will simply reduce some direct splatter, particularly back towards the faces of the dental staff, but given the well-known nature of dental aerosols there is no logical reason why these would be contained within a passive dome that has no air removal associated with it.

5.On line 82, figure 1 part A needs to explain which of the simulator units within the simulation laboratory was used, and clearly indicate this in part a of the figure and also in the legends of that figure.

6.On line 83 the description of the vertical and horizontal axes is confusing, since according to figure 1 part B these are behind the patient’s head and behind the operators head, however logically they should be towards the patient’s feet.

7.For line 86, describe how the working services were disinfected.

8.For line 104, state the nature of the insert used in the ultrasonic, and the type of bird that was used in the dental high-speed handpiece. Also specify what the water flow rate was to each of these devices (in mL/min).

9.For line 113, a wider angle photograph of the prime protector should be included to show how the 3 feet are located in relation to the dental chair.

10.For line 124, based on the non-normal distribution of the counts, either a non-parametric method should have been used for all analyses, or the CFU data log transformed before analysis. This is relevant to the analysis of results in line 141 were a parametric test was used.

11.The legend to table 1 needs to are the key values - they are not meeting differences as is stated in the title of table 1.

12.In the discussion on line 162, the authors should follow the hierarchy of risk controls, and list elimination, substitution, engineering controls, administrative controls and then PPE in that order from most effective to least effective. It is inappropriate to begin the discussion by listing barriers first.

13.For line 166, several clinical studies have now shown that the combination of preprocedural mouth rinsing and the use of high-volume evacuation allows dental practice to be conducted safely. These recent studies should be cited at this point, see DOI 10.1177/00220345211015948; DOI 10.1007/s00784-022-04532-8; and similar papers.

14.On line 201 it is stated that lactobacilli are Gram-negative - which is incorrect. While they could be described as facultative anaerobes it is technically more correct to describe them as being aerotolerant anaerobes.

15.On line 219, the inclusion of high-volume evacuation should be mentioned, since this measure is used routinely in modern dental practice, and it is known to be highly effective at mitigating the generation of aerosols and the spread of splatter into the operatory.

16.The authors could do a better job of discussing the limitations of their work, including the ways that their setup does not replicate what happens in a real working clinic.

17.On line 225 in the conclusion, the study did not test in any way the use of PPE so it is inappropriate to make commentary regarding the use of PPE.as well, PPE is the lowest of the risk control methods in terms of its hierarchy of use.

18.Several typographical areas are present

a.line 124 the word no should be non

b.line 134 the word stablished should be established

19.The references require revision

a. to ensure correct format to match the journal (including removing the name of the editor from references 1 and 18,

and

b, to ensure the correct use of capitals in the title of articles in references 3, 6, 7, 8, 12, 13, and 22.

20.Reference 8 is missing details of the journal name, journal volume and year of publication.

21.The supplied link for sharing data does not work.

22.The legend to figure 1 should state what the bars represent (median, 95% CI etc).

Reviewer #2: The manuscript entitled “Reduction of aerosol dissemination in a dental area generated by high-speed and scaler ultrasonic devices employing the “Prime Protector” to compare the spread of bioaerosols generated by a High-speed Handpiece (HH) and an Ultrasonic Prophylaxis Device (UPD), with and without the Prime Protector dome.

Some points raised:

1-The introduction is somehow poor and disorganized, there are several different devices that was developed and tested during the most problematic phase of pandemics, what is the difference showed here?

2-In the Introduction the authors stated that the previous studies “recent studies with different procedures and methodologies show a variety of results that leave gaps in evidence” what gap are that? Which one this paper solved? Is not clear. Besides, there are similar devices as the author present, included also in the reference list Montalli et al., 2021; Chestsuttayangkul et al.,

3-One of the major concern to use a device like Prime Protector is the disinfection needed after procedures made in one patient. There is no information regarding this concern. Different devices like the one described by Montalli (https://doi.org/

10.1371/journal.pone.0255533) using a metal support, with a 30 cm ring and covered by a disposable 30 microns thickness PVC film measuring approximately 1.5 x 1.5 m which is disposable. How was the cleaning and disinfection made? The same PP was used more than once? How the authors assure the complete disinfection? If PP was used more than once during the experiment, is it possible to have contamination from the previous experiment?

4-The methodology used should be referenced (http://dx.doi.org/10.1590/1981-863720200001820200088).

5-The table included in the text is not necessary since there is no relation with the manuscript.

6-Discussion should be deeper revised.

7-The legend of the Figure 3 is not clear.

8-There are some references with missing data in the list for example Chestsuttayangkul

6. PLOS authors have the option to publish the peer review history of their article (what does this mean?). If published, this will include your full peer review and any attached files.

Reviewer #1: **Yes: **Laurence Walsh

Reviewer #2: **Yes: **Marcelo Henrique Napimoga

---

## [Author Response · Author response to Decision Letter 0]

1 Jun 2023

Dear Rajeev Singh, 

Academic Editor

PLOS ONE

Based on the comments made by the academic reviewer, PLOS Editorial Office and reviewers, we proceed to answer each of the points.

For the PLOS Editorial Office. - We appreciate your recommendation to review and evaluate the specific previously published works and have taken them into account to improve our manuscript, as they are relevant to our investigation.

About the Journal Requirements. -

1. We have made sure that our revised manuscript meets PLOS ONE's style requirements. 

2. We have requested the architect Lissette Lucio (author of Figure 1) for her written permission to publish these figures specifically under the CC BY 4.0 license, through the completion of the Content Permission and uploaded that file as an "Other" file with our submission. 

To the reviewers, we considered it appropriate to resume the responses to their comments in the tables below.

To Reviewer #1: Laurence Walsh

We have considered your comments to our manuscript. The corrections change the lines’ numbers in the revised manuscript. 

1.For the abstract to be meaningful, it should state that there was no separate high-volume evacuation used, nor was there any air removal attached to the dome. It should also give the key dimensions of the dome such as its diameter and position in relation to the mouth. The correction regarding evacuation used can be found in the new lines 38 and 39. The correction regarding key dimensions of the dome can be found in the new lines 30-32.

2.On line 56, the accepted definition for aerosol particles is a size of less than 5 µm - not 50 µm. The correction can be found in the new line 61.

3.In the introduction on line 66, there is no mention of other effective risk reduction methods such as preprocedural use of antimicrobial mouth rinses. The correction can be found in the new lines 75-78

4.In line 68 some discussion about the use of the dome would be useful. Without any connected air removal the dome will simply reduce some direct splatter, particularly back towards the faces of the dental staff, but given the well-known nature of dental aerosols there is no logical reason why these would be contained within a passive dome that has no air removal associated with it. The correction can be found in the new lines 70- 74

5.On line 82, figure 1 part A needs to explain which of the simulator units within the simulation laboratory was used, and clearly indicate this in part a of the figure and also in the legends of that figure. Figure 1 and its legend was corrected.

6.On line 83 the description of the vertical and horizontal axes is confusing, since according to figure 1 part B these are behind the patient’s head and behind the operators head, however logically they should be towards the patient’s feet. Figure 1 part B was corrected .

7.For line 86, describe how the working services were disinfected. Description can be found in lines 111-113.

8.For line 104, state the nature of the insert used in the ultrasonic, and the type of bird that was used in the dental high-speed handpiece. Also specify what the water flow rate was to each of these devices (in mL/min). Ultrasonic and dental high-speed handpiece bur type can be found in new lines 128 and 129.Flow rate can be found in line 133

9.For line 113, a wider angle photograph of the prime protector should be included to show how the 3 feet are located in relation to the dental chai. Figure 2 was modified and a better description of the placement of the PP in lines 142-145

10.For line 124, based on the non-normal distribution of the counts, either a non-parametric method should have been used for all analyses, or the CFU data log transformed before analysis. This is relevant to the analysis of results in line 141 were a parametric test was used. The correction can be found in the new lines 156-158.

11.The legend to table 1 needs to are the key values - they are not meeting differences as is stated in the title of table 1. Table 1 was modified.

12.In the discussion on line 162, the authors should follow the hierarchy of risk controls, and list elimination, substitution, engineering controls, administrative controls and then PPE in that order from most effective to least effective. It is inappropriate to begin the discussion by listing barriers first. We followed this recommendation, and the revised version is on lines 200-212

13.For line 166, several clinical studies have now shown that the combination of preprocedural mouth rinsing and the use of high-volume evacuation allows dental practice to be conducted safely. These recent studies should be cited at this point, see DOI 10.1177/00220345211015948; DOI 10.1007/s00784-022-04532-8; and similar papers. We followed this recommendation, and the revised version is on lines 217-221

14.On line 201 it is stated that lactobacilli are Gram-negative - which is incorrect. While they could be described as facultative anaerobes it is technically more correct to describe them as being aerotolerant anaerobes. The correction can be found in the new line 255

15.On line 219, the inclusion of high-volume evacuation should be mentioned, since this measure is used routinely in modern dental practice, and it is known to be highly effective at mitigating the generation of aerosols and the spread of splatter into the operatory. The correction can be found in the new line 277

16.The authors could do a better job of discussing the limitations of their work, including the ways that their setup does not replicate what happens in a real working clinic. The correction can be found in the new lines 270-274

17.On line 225 in the conclusion, the study did not test in any way the use of PPE so it is inappropriate to make commentary regarding the use of PPE.as well, PPE is the lowest of the risk control methods in terms of its hierarchy of use. Usage of PPE was excluded from our conclusions as we didn’t test it

18.Several typographical areas are present

 a.line 124 the word no should be non

 b.line 134 the word stablished should be established 

The correction can be found in the new lines 156 and 161.

19.The references require revision

 a. to ensure correct format to match the journal (including removing the name of the editor from references 1 and 18, and b, to ensure the correct use of capitals in the title of articles in references 3, 6, 7, 8, 12, 13, and 22. a. All editors from the references were eliminated.

b. The titles of the mentioned articles were checked and corrected

20.Reference 8 is missing details of the journal name, journal volume and year of publication. Previous reference 8 is now reference 9, which has been corrected.

21.The supplied link for sharing data does not work. We have fixed this problem. Files can be found in the new link: https://drive.google.com/drive/folders/1mNxcsK_XNFPWAPoNSE1VvjpMYCR1qdL1

22.The legend to figure 1 should state what the bars represent (median, 95% CI etc). Figure 1 did not have bars. Figure 3 had bars and its legend was corrected following this comment.

 To reviewer #2: Marcelo Henrique Napimoga

We have considered your comments to our manuscript. The corrections change the lines’ numbers in the revised manuscript.

1-The introduction is somehow poor and disorganized, there are several different devices that was developed and tested during the most problematic phase of pandemics, what is the difference showed here? The introduction was revised, and it changed all the manuscript’s line numbers. The difference between pre and post pandemic protective devices is established in line. We hope your concerns have been solved.

2-In the Introduction the authors stated that the previous studies “recent studies with different procedures and methodologies show a variety of results that leave gaps in evidence” what gap are that? Which one this paper solved? Is not clear. Besides, there are similar devices as the author present, included also in the reference list Montalli et al., 2021; Chestsuttayangkul et al. We have corrected the introduction and the contribution of our study is from line 79 to 88.

3-One of the major concern to use a device like Prime Protector is the disinfection needed after procedures made in one patient. There is no information regarding this concern. Different devices like the one described by Montalli (https://doi.org/ 10.1371/journal.pone.0255533) using a metal support, with a 30 cm ring and covered by a disposable 30 microns thickness PVC film measuring approximately 1.5 x 1.5 m which is disposable. How was the cleaning and disinfection made? The same PP was used more than once? How the authors assure the complete disinfection? If PP was used more than once during the experiment, is it possible to have contamination from the previous experiment? Changes regarding the disinfection process can be found in lines 111-113 and the disinfection protocol is now uploaded as Supporting Information

4-The methodology used should be referenced (http://dx.doi.org/10.1590/1981-863720200001820200088).

The correction can be found in the new lines 90-94

5-The table included in the text is not necessary since there is no relation with the manuscript. The table mentioned was eliminated

6-Discussion should be deeper revised. We have added and edited the discussion following the recommendations from reviewer #1

7-The legend of the Figure 3 is not clear. The legend of figure 3 was corrected

8-There are some references with missing data in the list for example Chestsuttayangkul The references were revised and updated.

We hope to have met all the requirements and we are open to answering any other questions or concerns.

Thanks for your gentle attention.

Sincerely, 

Dr. Carolina Guzmán

Correponding author

---

## [Decision Letter · Decision Letter 1]

3 Jul 2023

Reduction of aerosol dissemination in a dental area generated by high-speed and scaler ultrasonic devices employing the “Prime Protector"

PONE-D-22-32218R1

Dear Dr. Guzmán Flores,

We’re pleased to inform you that your manuscript has been judged scientifically suitable for publication and will be formally accepted for publication once it meets all outstanding technical requirements.

Kind regards,

Rajeev Singh

Academic Editor

PLOS ONE

Additional Editor Comments (optional):

Reviewers' comments:

Reviewer's Responses to Questions

**Comments to the Author**

1. If the authors have adequately addressed your comments raised in a previous round of review and you feel that this manuscript is now acceptable for publication, you may indicate that here to bypass the “Comments to the Author” section, enter your conflict of interest statement in the “Confidential to Editor” section, and submit your "Accept" recommendation.

Reviewer #1: All comments have been addressed

2. Is the manuscript technically sound, and do the data support the conclusions?

Reviewer #1: Yes

3. Has the statistical analysis been performed appropriately and rigorously? 

Reviewer #1: Yes

4. Have the authors made all data underlying the findings in their manuscript fully available?

Reviewer #1: Yes

5. Is the manuscript presented in an intelligible fashion and written in standard English?

Reviewer #1: Yes

6. Review Comments to the Author

Reviewer #1: All concerns have been addressed in this revision and the paper now reads very well. The authors provided a detailed response and a tracked changes version showing all the changes that were made.

7. PLOS authors have the option to publish the peer review history of their article (what does this mean?). If published, this will include your full peer review and any attached files.

Reviewer #1: **Yes: **Laurence J Walsh

---

## [Editor Report · Acceptance letter]

27 Jul 2023

PONE-D-22-32218R1 

Reduction of aerosol dissemination in a dental area generated by high-speed and scaler ultrasonic devices employing the “Prime Protector” 

Dear Dr. Guzmán-Flores:

I'm pleased to inform you that your manuscript has been deemed suitable for publication in PLOS ONE. Congratulations! Your manuscript is now with our production department. 

Kind regards, 

on behalf of

Dr. Rajeev Singh 

Academic Editor

PLOS ONE